# In Situ Cell Signalling of the Hippo-YAP/TAZ Pathway in Reaction to Complex Dynamic Loading in an Intervertebral Disc Organ Culture

**DOI:** 10.3390/ijms222413641

**Published:** 2021-12-20

**Authors:** Andreas S. Croft, Ysaline Roth, Katharina A. C. Oswald, Slavko Ćorluka, Paola Bermudez-Lekerika, Benjamin Gantenbein

**Affiliations:** 1Tissue Engineering for Orthopaedics and Mechanobiology, Bone & Joint Program, Department for BioMedical Research (DBMR), Medical Faculty, University of Bern, CH-3008 Bern, Switzerland; andreas.croft@dbmr.unibe.ch (A.S.C.); ysaline.roth@students.unibe.ch (Y.R.); katharina.oswald@insel.ch (K.A.C.O.); slavko.corluka@insel.ch (S.Ć.); paola.bermudez@dbmr.unibe.ch (P.B.-L.); 2Department of Orthopaedic Surgery and Traumatology, Inselspital, Bern University Hospital, University of Bern, CH-3010 Bern, Switzerland

**Keywords:** Hippo-YAP/TAZ pathway, intervertebral disc degeneration, complex dynamic loading, organ culture, bioreactor

## Abstract

Recently, a dysregulation of the Hippo-YAP/TAZ pathway has been correlated with intervertebral disc (IVD) degeneration (IDD), as it plays a key role in cell survival, tissue regeneration, and mechanical stress. We aimed to investigate the influence of different mechanical loading regimes, i.e., under compression and torsion, on the induction and progression of IDD and its association with the Hippo-YAP/TAZ pathway. Therefore, bovine IVDs were assigned to one of four different static or complex dynamic loading regimes: (i) static, (ii) “low-stress”, (iii) “intermediate-stress”, and (iv) “high-stress” regime using a bioreactor. After one week of loading, a significant loss of relative IVD height was observed in the intermediate- and high-stress regimes. Furthermore, the high-stress regime showed a significantly lower cell viability and a significant decrease in glycosaminoglycan content in the tissue. Finally, the mechanosensitive gene *CILP* was significantly downregulated overall, and the Hippo-pathway gene *MST1* was significantly upregulated in the high-stress regime. This study demonstrates that excessive torsion combined with compression leads to key features of IDD. However, the results indicated no clear correlation between the degree of IDD and a subsequent inactivation of the Hippo-YAP/TAZ pathway as a means of regenerating the IVD.

## 1. Introduction

Intervertebral disc (IVD) degeneration (IDD) is a leading cause of chronic low back pain (LBP) [1]. The exact aetiology of IDD is still unknown but the origins of the disease are believed to be multifactorial [2]. Risk factors that may favour the initiation and progression of IDD include trauma [3], smoking [4,5], genetic factors [6,7], and mechanical stress [8,9]. It is assumed that a close interplay between mechanical overloading, catabolic cell response and the degeneration of the IVD’s extracellular matrix (ECM) form a vicious circle that leads to the inevitable progression of IDD [2]. The first signs of IDD are usually observed in the IVD’s core tissue, known as the nucleus pulposus (NP) [10]. Changes in the anabolic and catabolic turnover cause an excessive breakdown of hydrophilic proteoglycans and glycosaminoglycans (GAG) and consequently lead to the dehydration of the NP’s highly hydrated nature [11]. As a result, the expansive force of the NP decreases, IVD height is lost, and the compressive stress shifts from the NP to the surrounding fibrous tissue called the annulus fibrosus (AF) [12,13]. Increased biomechanical “wear and tear” of the AF and altered stress distribution then accelerates its degenerative progression, which is characterised by the presence of tears and fissures within the AF as well as by vascularisation and innervation [14,15]. Consequently, the IVD becomes even less resistant to mechanical stress and more susceptible to damage [16].

The Hippo-YAP/TAZ pathway (yes-associated protein 1 and transcriptional coactivator with PDZ-binding motif, respectively) is an evolutionarily conserved signalling cascade that was first discovered in *Drosophila melanogaster* and later confirmed in mammals [17,18]. The pathway plays a crucial role in tissue homeostasis, organogenesis, and tumorigenesis by controlling cell proliferation and apoptosis [19]. YAP/TAZ are transcriptional coactivators, and both function as negative regulators of the Hippo-YAP/TAZ pathway [20]. Once the pathway is activated, mammalian STE20-like protein kinases 1 and 2 (MST1/2) autophosphorylate, form heterodimers with the Salvador family WW-domain-containing protein 1 (SAV1) and subsequently phosphorylate large tumor suppressor kinases 1 and 2 (LATS1/2) [21,22]. In conjunction with mob kinase activator 1 (MOB1), LATS1/2 then directly phosphorylate YAP/TAZ, thereby inhibiting their nuclear translocation and inhibiting cell proliferation and tissue growth [19,23]. However, in the presence of specific growth factors like the MAP/microtubule affinity-regulating kinase 4 (MARK4) [24], cell attachment [25], altered cell polarity [26], and mechanical stimuli, i.e., shear forces and tissue stretching [27], the Hippo-YAP/TAZ pathway is deactivated. Hence, YAP/TAZ are dephosphorylated and translocate into the nucleus, where they bind to the TEA-domain (TEAD) transcription family to promote transcriptional programs important for cell survival, migration, and proliferation [28].

In the past, complex dynamic loading of bovine IVDs cultured ex vivo has been shown to induce clear evidence of IDD [29]. However, it is not known if and how the Hippo-YAP/TAZ pathway responds to complex dynamic loading. Therefore, the aim of this study was to investigate the influence of (i) static and (ii) different complex dynamic loading regimes, more specifically of combined compression and torsion, on the IVD and its endogenous regulation of the Hippo-YAP/TAZ pathway using a bovine explant culture model. To execute combined compression and torsion, a custom-built bioreactor that enables two degrees-of-freedom movement was utilised. We hypothesised that more intense loading regimes would promote more pronounced degrees of IDD and consequently lead to alterations in the Hippo-YAP/TAZ signalling cascade.

## 2. Results

### 2.1. IVD Height Changes

After one week in the bioreactor, a significant loss (*p* < 0.0001) of the absolute height was recorded in every mechanical loading regime tested (Figure 1a). To evaluate the height changes caused by each loading regime more precisely, the IVDs’ height loss was calculated relative to the initial height after their isolation from the bovine tail. That way, the intermediate- (*p* < 0.0001) and high-stress (*p* < 0.01) regimes both showed a significant relative reduction in the IVDs’ height (intermediate-stress: 26.1 ± 4.3% reduction and high-stress: 20.9 ± 1.7% reduction, respectively), and a strong trend was observed with the low-stress regime (*p* = 0.06) (Figure 1b). Moreover, the intermediate-stress regime was significantly more affected by a relative decrease in the IVDs’ height than the static load regime (*p* < 0.05).

### 2.2. Cell Viability

To further assess the influence of the different loading regimes on the tissue’s health, the cell viability was analysed. Both the NP and the AF tissues showed a correlation between the intensity of the loading regimes and the extent of cell death. Generally, increasingly intense loading regimes also lead to an increase in cell death (Figure 1c). No living cells were detected within the NP when the high-stress regime was applied. The cell viability was significantly decreased compared to day 0 (*p* < 0.01), the static load (*p* < 0.05) and the low-stress (*p* < 0.05) regime. The same tendency was found within the AF, however, the high-stress regime proved to be less detrimental for the AF than to the NP (16.1 ± 11.3% cell viability for the AF). Furthermore, the cell viability was significantly decreased in the high-stress regime compared to day 0 (*p* < 0.05) and a strong trend was observed between the static load regime and the high-stress regime for the AF (*p* = 0.057).

### 2.3. Glycosaminoglycan Content

The measured amount of glycosaminoglycans (GAG) in the NP and AF tissue was put in relation to the dry weight as well as to the amount of DNA of the respective isolated tissue. In the NP, the high-stress regime caused a significant decrease of GAG per tissue dry weight (*p* < 0.01 compared to the intermediate-stress regime and *p* < 0.05 compared to the other regimes) (Figure 2a). However, all the other regimes did not significantly differ from each other in the NP as well as all regimes regarding the AF were comparable. Despite these significant differences in the tissue dry weight, the same observation could not be confirmed when calculating the GAG/DNA ratio. Only the difference in GAG between the high-stress regime and the static load regime in the NP showed a similar trend (*p* = 0.15) to that of the tissue dry weight (Figure 2b).

As for the GAG released into the culture medium, a stress-dependent effect for the complex dynamic loading regimes was noted. The higher the intensity of the loading regime, the more GAG was released into the medium. Moreover, the IVDs in the high-stress regime released significantly (*p* < 0.05) more GAG into the culture medium than the day 0 samples (Figure 2c).

### 2.4. Nitric Oide Content

The loading regime showed to have a similar influence on the release of nitric oxide (NO) from the IVDs into the medium as it had on the release of GAG. Hence, the stress-dependent effect that was provoked by the different complex dynamic loading regimes could be confirmed with the NO measurements. In particular, a trend (*p* = 0.15) of gradually increasing NO levels was recorded between the low-stress and the high-stress regime (Figure 2d).

### 2.5. Gene Expression

To determine the influence of static or complex dynamic loading regimes on the gene expression of NP and AF tissue, genes considered as key players of the Hippo-YAP/TAZ pathway (*YAP*, *TAZ*, *LATS1,* and *MST1*) IVD-related markers of ECM synthesis (*ACAN*, *COL1,* and *COL2*), ECM degradation (*ADAMTS4*, *MMP13,* and *MMP3*), mechanosensitivity (*COMP* and *CILP*), as well as the presence of inflammation markers (*COX2*, *MCP1,* and *RANTES*) were analysed.

Concerning the Hippo-YAP/TAZ pathway, a gradual, but non-significant, stress-dependent decrease of *YAP* was noticed in the NP. The less intense the complex dynamic loading regime, the more *YAP* was downregulated. Furthermore, *MST1* was significantly (*p* < 0.05) upregulated in the high-stress regime compared to the low-stress regime in the NP (Figure 3a). However, these findings could not be confirmed in the AF (Figure 3b). The relative gene expression of the anabolic marker *COL1* revealed a significant upregulation in the intermediate-stress regime compared to the static load (*p* < 0.05) and the low-stress regime (*p* < 0.05) in the NP (Figure 3c), but no changes were found for any anabolic marker in the AF (Figure 3d). Then, the catabolic genes *ADAMTS4* and *MMP13* were generally, though non-significantly, upregulated in the NP and the AF. In the NP, the highest peaks were found in the intermediate-stress regime (3528 ± 7804-fold upregulation for *ADAMTS4* and 3307 ± 7150-fold upregulation for *MMP13*) (Figure 3e) and in the AF, all regimes showed similar high upregulations for *ADAMTS4* and *MMP13* (up to 682 ± 1104-fold upregulation) (Figure 3f). As for the catabolic genes, the inflammation-related genes *COX2* and *MCP1* were generally, but not significantly, upregulated, especially for the intermediate- (44.7 ± 48.8-fold upregulation for *MCP1*) and high-stress regime (19 ± 23-fold upregulation for *MCP1*) in the NP (Figure 3g). No differences were again observed in the AF (Figure 3h). Finally, the mechanosensitive maker *COMP* was significantly (*p* < 0.05) downregulated (0.21 ± 0.08-fold downregulation) in the NP compared to day 0 and the relative gene expression of *CILP* was significantly lower than day 0 for every regime measured (static load: 0.08 ± 0.12-fold downregulated, low-stress: 0.24 ± 0.5-fold downregulated, intermediate-stress: 0.23 ± 0.35-fold downregulated, and high-stress: 0.15 ± 0.22-fold downregulated) (Figure 3i). However, this could not be confirmed in the AF (Figure 3j).

### 2.6. Immunofluorescence of Nuclear Translocated YAP

Immunofluorescence pictures were taken to see if different loading regimes would lead to visible changes of dephosphorylated/active YAP in the cells’ nuclei. First of all, dephosphorylated YAP was mainly found in the cells’ nuclei. Extranuclear YAP was only observed in the static load (Figure 4b_3) and low-stress regimes (Figure 4c_3) in the NP and sporadically in the day 0 regime in the AF (Figure 4f_3). Furthermore, based on visual examination, the intensity of YAP present in the cells’ nuclei was very comparable across the different loading regimes for the NP (Figure 4a_2–4e_2) and the AF (Figure 4f_2–4g_2). Finally, the amount YAP-positive nuclei relative to the number of nuclei per regime was very similar throughout the NP tissue (day 0: 61.3%, static load: 60.5%, low-stress: 55.6%, intermediate-stress: 56%, and high-stress: 57.1% YAP-positive, respectively; Figure 4a_2–4e_2) but underrepresented in the AF’s day 0 (50% YAP-positive; Figure 4f_3) and low-stress regime (47.9% YAP-positive; Figure 4h_3) compared to the other regimes in the AF (static load: 78.9%, intermediate-stress: 67.9% and high-stress: 77% YAP-positive, respectively) (close up pictures can be found in Appendix A).

## 3. Discussion

In this study, we showed that more intense complex dynamic loading regimes lead to more pronounced levels of IDD. More specifically, the intermediate- and high-stress regimes promoted changes in the IVD, which are seen as key features of IDD. A first indication of IDD was delivered by the changes of the IVD’s absolute height, which were significantly lower in all regimes compared to their initial height after removal from the bovine tail. Even though reversible height changes of 8% in human IVDs [30] and up to 10% in bovine IVD organ culture models [31] are considered as physiological, the values observed here clearly exceeded the physiological range with significant relative height losses of more than 20% in the intermediate- and high-stress regimes. Lang et al. also utilised a bovine organ culture model using a bioreactor and stated a decrease of 20% in the IVD’s height was pathological [31]. Furthermore, a recent study showed that a reduction in the IVD’s height caused by a pathological loading regime correlated with a significant upregulation of proinflammatory cytokines like IL-6 and IL-8, indicating the onset of IDD [32]. With the results at hand and the fact that a permanent decrease of the IVD’s height is strongly correlated with IDD [10], we can already conclude that there is an association between more intense complex loading regimes and IDD. Similar studies on the effect of high mechanical impact and acute mechanical injury were carried out using micro fractures and/or the impact on the endplates [33,34,35]. These studies all reported the release of proinflammatory cytokines such as IL-6 and IL-8 and stress-factors in relation to mechanical overloading. However, none of them were able to investigate torsion as destructive movement.

Moreover, further evidence for the presence of IDD due to torsion-overloading was found when analysing the cell viability. In the NP, the high-stress regime was responsible for a significant decrease in cell viability compared to all regimes except the intermediate-stress regime, and in the AF, the viability was significantly lower than on day 0. In the past, physical overloading of the IVD has been shown to be associated with increased cell death and ECM degradation and consequently with the initiation and progression of IDD [36]. For example, Zhou et al. overloaded bovine IVDs using a one strike organ culture model to compress the IVDs to 50% of their initial height [37]. Compared to the control group, severe compression led to a significantly lower cell viability in the NP and the AF. Similar results concerning the cell viability were also obtained in a bovine disc organ culture study by Chan et al., where the combined effect of cyclic compression and 2° torsion was tested over a period of 14 days [29]. Surprisingly, only the NP suffered from extensive cell death with cell survival rates less than 10%, while the AF was not affected. Cell viability was also significantly affected in a “wedged-loaded” asymmetric loading system [38].

Another central discovery that is typical for IDD was the significant reduction of the NP’s GAG content in the high-stress regime. Since a minimal amount of undegraded GAG is crucial for the NP’s homeostasis, a loss of GAG will lead to tissue dehydration and ultimately to its degeneration [39,40]. Interestingly, the high-stress regime also evoked an increased release of GAG into the culture medium compared to the day 0, thus funding the hypothesis that a part of the GAG diffused from the NP into the medium. Moreover, the comparable tendency of NO production and its release into the culture medium reinforced the findings observed with the GAG in the medium. Similarly to the GAG content, a trend of increased NO production in the high-stress regime compared to the low-stress regime or on day 0 was observed. The production of NO is known to be triggered by hydrostatic pressure or increased shear stress [41,42]. Furthermore, increased NO levels have been associated with decreased proteoglycan production and increased apoptosis in the IVD [41,43].

Further evidence for IDD was then found at the gene level. In the NP, *COL1* was significantly upregulated in the intermediate-stress regime compared to the static load and the low-stress regime. An upregulation of *COL1* is seen as an indication that the NP cells dedifferentiated more towards a fibroblast-like phenotype. This is a condition that is frequently seen during extensive shear stress and IDD in general [2,44]. Although none of the catabolic genes or inflammatory genes showed significant changes, they were generally upregulated, especially for the intermediate- and high-stress regimes. This was most notably evident with *MCP1*, *MMP13,* and *ADAMTS4* in the NP, but also visible with *RANTES*. Given that MCP1 is viewed as a critical factor for the initiation of an inflammatory response in the IVD [45] and that an imbalanced upregulation of matrix remodelling genes is associated with IDD [46,47], these findings lead to the assumption that complex dynamic loading can start as well as aggravate an inflammatory and degenerative process in the IVD.

Although we could show that complex dynamic loading, particularly the intermediate- and the high-stress regimes, overloaded the IVDs and consequently caused IDD, finding a correlation between IDD and the Hippo-YAP/TAZ pathway was more ambiguous. The immunofluorescence pictures revealed that YAP managed to translocate into the cells’ nucleus. Extranuclear YAP was only observed in the less intense loading regimes, i.e., in the static load and low-stress regimes in the NP and sporadically in the day 0 regime in the AF. This nuclear localisation is mainly observed when the Hippo-YAP/TAZ pathway is inactivated, which results in the activation of YAP [22]. Increased activation and subsequent nuclear localisation of YAP with more intense loading regimes would confirm previous research showing that mechanical stress can trigger the translocation of YAP and TAZ to promote gene expression [48,49]. At gene level, however, no significant upregulation of *YAP* or *TAZ* was found and instead a significant increase in *MST1* expression was detected for the high-stress regime in the NP. Interestingly, MST1 is known to be activated when the Hippo-YAP/TAZ pathway is also active, or in other words when YAP is mostly inactive [20]. An explanation for this significant upregulation of *MST1* in the NP and the unchanged or non-significant downregulation of *YAP* and *TAZ* could result when considering the mechanosensitive markers *COMP* and *CILP*. In the NP, *COMP* was significantly downregulated in the static load regime and to our surprise, *CILP* was significantly lower in every single loading regime compared to day 0, but most prominently again in the static load regime. In the past, studies have shown a correlation between mechanical stress and an increase of COMP and CILP [50,51,52]. What these studies have in common, however, is the fact that all the samples tested were always in motion, at least during their “active phase”. In contrast, the IVDs in our study were only in motion when the torsion was activated, so just for four hours during the “active phase”. During the remaining four hours of the “active phase”, the IVDs were exposed to a higher, but static, pressure. It is therefore possible that the IVDs were either mechanically not stimulated enough or that their stimulation varied too little. The latter, however, seems more likely as the IVDs clearly displayed a degenerated phenotype, which was almost certainly induced by overloading the IVDs. In summary, we assumed that the Hippo-YAP/TAZ pathway was predominantly activated due to a lack of movement, which led to the inactivation of YAP and TAZ and consequently prevented them from initiating transcriptional programs to induce cell survival and tissue regeneration [22].

A similar study on the influence of different hydrostatic pressure intensities on the endogenous YAP regulation and the ECM turnover of rabbit IVDs was carried out by Wang et al. [53]. There, significant differences in the proteomic YAP expression were only found at a minimal pressure of 0.8 MPa in the NP and 1.0 MPa in the AF, both of which are higher than any of the pressures that were applied in our study. Furthermore, in the quoted study, only the impact of pressure on the IVD was analysed, without the addition of torsion.

Nevertheless, whether and to which extent the Hippo-YAP/TAZ pathway is involved in IDD and IVD regeneration is still elusive, and recent findings on this topic are not always coherent. For example, Zhang et al. [54] showed a significant overexpression of YAP after acute disc injury in rats. Surprisingly, however, YAP levels gradually decreased as healthy rats aged and naturally caused IDD became apparent. A follow-up study by the same research group confirmed this gradual decrease of YAP with increasing age and natural IDD [55]. Although YAP levels were increased after disc injury, these elevated levels could not protect the IVD from further degeneration [55].

As a result, further research is needed to establish and prove a given relationship between the Hippo-YAP/TAZ pathway and IDD, and if there is a clear link, then how and to what extent this pathway could be used to inhibit IDD or even to regenerate a degenerated or damaged IVD.

Finally, there are limitations to this study that need to be addressed. Firstly, as previously discussed, there was no loading regime that used a sine wave function to control the compression. Therefore, we cannot state how the IVDs and more specifically the Hippo-YAP/TAZ pathway would have reacted if the samples had been exposed to a constantly varying pressure. Secondly, only activated/non-phosphorylated YAP was evaluated for the immunohistochemistry assay. The addition of an antibody specifically for phosphorylated YAP and its subsequent visualisation would have made it possible to determine the ratio between activated and deactivated YAP per cell and consequently what percentage of the total YAP translocated into the cell’s nucleus.

## 4. Materials and Methods

### 4.1. Isolation of Bovine IVDs and Organ Culture

Fresh and complete caudal IVDs were collected from approximately one year old animals, which had been sacrificed three to four hours before. For superficial disinfection, the bovine tail was first immersed into 1% Betadine^®^ solution (Mundipharma Medical Company, Basel, Switzerland). After five minutes, the tail was moved onto a sterile cutting board, where the tendons and muscle tissue surrounding the IVDs were removed using a sterile scalpel blade. After clearing the surrounding tissue, the IVDs were dissected from the vertebral bodies using a custom-made industrial blade holder and then wrapped in sterile gauze, which had been soaked in 0.9% sodium chloride and 55 mM sodium citrate. As soon as five IVDs per bovine tail were isolated, the discs’ diameters were measured twice with a calliper at positions offset by 90°. Then the IVDs’ heights were measured from one CEP to the other CEP, and it was ensured that the CEPs were aligned horizontally between the jaws of the calliper as previously described [56]. In a next step, all IVDs were rinsed with Ringer’s lactate solution (#FE1010206; Bichsel, Interlaken, Switzerland) using a Zimmer Pulsavac PlusTM jet-lavage system (Zimmer Biomet, Inc., Winterthur, Switzerland). Then, each IVD was twice fully immersed in penicillin/streptomycin (1000 U/mL; #5711; Sigma-Aldrich, Buchs, Switzerland) for five minutes and then rinsed with phosphate-buffered salt solution (PBS). After sterilisation, the IVDs were transferred separately into a beaker filled with high-glucose (4.5 g/L) Dulbecco’s Modified Eagle Medium (HG-DMEM; #52100-039; Gibco, Life Technologies, Zug, Switzerland), supplemented with 5% foetal bovine serum (FBS; #F7524; Sigma-Aldrich), 0.22% sodium hydrogen carbonate (#31437-500G-R; Sigma-Aldrich), 10 mM HEPES buffer solution (#15630-056; Thermo Fisher Scientific, Basel, Switzerland), 1 mM sodium pyruvate (#11360-039; Thermo Fisher Scientific), and penicillin/streptomycin/glutamine (100 U/mL, 100 and 292 μg/mL, respectively; #10378-016; Thermo Fisher Scientific). In this “free swelling” phase, all samples were incubated at 37 °C in normoxia and 5% CO_2_ for two days.

After surpassing these two days, one randomly picked IVD was assigned to the “day 0” regime and directly processed for further analysis. The remaining IVDs were then randomly assigned to one of four different static or complex dynamic mechanical loading regimes for seven days, i.e., (i) “static load”, (ii) “low-stress”, (iii) “intermediate-stress”, and (iv) “high-stress”, using a bioreactor that allows two Degree-of-Freedom (2DoF) loading (Table 1). All IVDs were cultured in enriched HG-DMEM at 37 °C with 20% O_2_ and 5% CO_2_, while medium change was performed twice a week.

The IVDs assigned to the static loading regime were exposed to a constant basal pressure of 0.1 MPa. In contrast, the IVDs assigned to the complex dynamic loading regimes underwent a passive phase of loading with a basal pressure of 0.1 MPa for 16 h, followed by an active phase that lasted eight hours. During this active phase, the IVDs went through a repeating sequence consisting of active pressure for one hour followed by torsion under basal pressure for another hour. This cycle of passive phase followed by an active phase was repeated every 24 h and was considered to mimic the diurnal rhythm of mechanical loading to which the IVDs are physiologically exposed in the spine (Figure 5).

### 4.2. Cell Viability

To assess the cell viability of each regime, NP and AF tissue samples were dissected into approximately 3 mm^3^ pieces. The tissue pieces were then immersed into serum-free medium containing 5 µM calcein-AM (#17783-1MG; Sigma-Aldrich) to stain the living cells and 1 µM ethidium homodimer (#46043-1MG-F; Sigma-Aldrich) to stain the dead cells and incubated at 37 °C. After two hours of incubation, 3D stacked images were taken on a confocal laser scanning microscope (cLSM710; Carl Zeiss; Jena, Germany) at 10× magnification. Finally, the living and dead cells were quantified using a custom-made macro for ImageJ software version 1.52 [57].

### 4.3. Glycosaminoglycan (GAG) Content

NP and AF tissue were isolated from the IVDs using an 8 mm biopsy punch (#05.275.38; Kai Medical, Seki, Japan, distributed by Polymed Medical Center, Glattbrugg, Switzerland) and a #10 scalpel blade (#CE20.1; Carl Roth Inc.; Karlsruhe, Germany). Then, approximately one third of each tissue sample was dried overnight at 60 °C. On the following day, the tissue dry weight was determined, and all samples were subsequently digested overnight in a 5 mM L-cysteine hydrochloride (#20119; Sigma-Aldrich) enriched papain solution (3.9 U/mL; #P3125; Sigma-Aldrich) at 60 °C. To quantify the sulphated GAG, 1,9-dimethyl-methylene blue zinc chloride double salt dye (#341088; Sigma-Aldrich) was applied to the digested tissue and the culture medium. The absorbance was then read at 600 nm using an ELISA reader (Spectramax M5, Molecular Devices, distributed by Bucher Biotec, Basel, Switzerland). Chondroitin sulfate sodium salt from bovine cartilage (#6737, Sigma-Aldrich) served as a standard to calculate the GAG concentration. In the end, the amount of GAG in the tissue was put in relation either to the dry weight of the isolated tissue or to the amount of DNA in the tissue. The GAG content that was released into the culture medium was normalised to the IVD volume after its isolation from the bovine tail.

### 4.4. DNA Content

The DNA content was determined from papain digested tissue using Hoechst 33,258 dye (#86d1405; Sigma-Aldrich). The fluorescence was measured at a wavelength of 350 nm excitation and 450 nm emission and the DNA concentration was calculated using a standard curve obtained with DNA sodium salt from calf thymus (#D1501; Sigma-Aldrich).

### 4.5. Nitric Oxide (NO) Content

The amount of NO radicals released by the IVDs was indirectly quantified by Griess reaction that measures the concentration of the more stable nitrite (NO_2_^−^). Therefore, the culture medium was collected on the last day of organ culture and then underwent protein precipitation with 70% ethanol. In a next step, the supernatant was mixed with N-(1-naphthyl)ethylenediamine dihydrochloride (#N9125; Sigma-Aldrich), sulfanilamide (#S9251; Sigma-Aldrich), and 10% phosphoric acid (#79617; Sigma-Aldrich). Then, the absorbance was read at a wavelength of 530 nm for the actual measurement and additionally at 650 nm to exclude the presence of any unspecific absorbance. Finally, the NO content was normalised to the IVD volume after its isolation from the bovine tail.

### 4.6. RNA Extraction and Relative Gene Expression

Approximately two thirds of the isolated NP and AF tissue was snap-frozen in liquid nitrogen. The frozen tissue was then pulverised into a powder using a mortar and transferred into 1 mL TRIzol reagent^®^ (#TR118; Molecular Research Center; Cincinnati, OH, USA, distributed by Lucerna-Chem inc., Lucerne, Switzerland) mixed together with 5 µL polyacryl carrier (#PC152; Molecular Research Center). Next, 1-Brome-3-Chloropropane (BCP; #B9673; Sigma-Aldrich) was used for phase separation. While AF samples were directly put in molecular grade absolute ethanol (#51976; Sigma-Aldrich), the NP samples first underwent a precipitation step to remove the residual GAG. Therefore, the RNA was precipitated with molecular grade isopropanol (#I9516; Sigma-Aldrich) and high salt precipitation solution (#PS161; Molecular Research Center).

In a next step, the RNA from both tissue samples was isolated using a GenElute miniprep kit (#RTN70; Sigma-Aldrich) and the remaining DNA was digested with an On-Column DNase I Digestion Set (#DNASE70; Sigma-Aldrich) following the user manual of the respective kit. The isolated RNA was then transformed to cDNA with a High Capacity cDNA Reverse Transcription kit (#4368814; Thermo Fisher Scientific) and a MyCycler™ Thermal Cycler system (#1709703; Bio-Rad Laboratories; Cressier, Switzerland). Afterwards, the cDNA was mixed with iTaq Univeral SYBR Green Supermix (#1725122; Bio-Rad) and with the primers of interest (Table 2). Finally, quantitative polymerase chain reaction (qPCR) was performed on a CFX96™ Real-Time System (#185-5096; Bio-Rad Laboratories). The relative expression was determined using the 2^−ΔΔCt^-method [58] while the ribosomal 18S RNA gene was used as a reference.

### 4.7. Immunofluorescence

IVDs were first halved sagittally and then fixed in 4% paraformaldehyde (PFA) (#1.04005.1000; Fluka; Buchs, Switzerland). After 50 h of fixation, the PFA was rinsed out and the IVDs were decalcified for five weeks in 12.5% EDTA (#03685; Fluka). In a next step, the IVDs were embedded in paraffin and then cut into 5 µm thin sections with a microtome (Microm HM355; Thermo Fisher Scientific).

After dewaxing, the antigens were unmasked using 0.1% Difco^TM^ trypsin (#215240; BD Biosciences; San Jose, CA, USA) at 37 °C for 20 min. Then, the samples were incubated at 4 °C overnight with anti-YAP1 (1:100; #bs-3605R; Bioss Antibodies Inc.; Woburn, MA, USA), which served as a primary antibody. After washing, polyclonal goat anti-rabbit IgG Alexa Fluor 488 (1:200; #A11008; Thermo Fisher Scientific) was used as a secondary antibody, with which the samples were incubated for one hour at room temperature in the dark. Then, all samples were mounted using fluoroshield mounting medium with DAPI (#ab104139; Abcam; Cambridge, United Kingdom, distributed by Lucerna-Chem, inc., Lucerne, Switzerland) and the images were subsequently obtained using a confocal laser scanning microscope (sLSM710; Carl Zeiss, Jena, Germany). Finally, YAP-positive nuclei were counted manually using the ImageJ cell counter plugin (version 1.52q for Mac OS X).

### 4.8. Statistics

For all quantitative data, a nonparametric distribution was assumed. The data are presented as mean ± standard deviation (SD), except for the absolute height, where just the mean is shown. The relative gene expression, the cell viability, the GAG per DNA content, and the GAG per tissue dry weight quantification were analysed by two-way ANOVA followed by Tukey’s multiple comparisons test. A Kruskal–Wallis test followed by Dunn’s multiple comparisons test was performed for the relative IVD height, the GAG in the culture medium and the NO content. Finally, the absolute IVD height was analysed by Repeated Measures (RM) two-way ANOVA followed by Sidak’s multiple comparisons test. All the tests were performed using GraphPad Prism (version 7.0e for Mac OS X, GraphPad Software; San Diego, CA, USA) and a *p*-value < 0.05 was considered statistically significant. Up to seven replicates were used for each experiment, however, the exact number of biological replicates = N and/or technical replicates = n are indicated in each figure legend.

## 5. Conclusions

This study shows that mechanically intense loading regimes using complex dynamic loading overload the IVD and lead to key features of IDD. The intensity of the loading regime used and the consequent degree of IDD, however, cannot be clearly associated with an inactivation of the Hippo-YAP/TAZ pathway as a means of promoting cell survival and tissue regeneration.

Clinically, the study addresses the link of hyperphysiological range of torsion and its effect on IVD health and increased pain. This correlation has been identified in clinical studies using patients with LBP and increased torsional range of motion [59]. Here, we showed evidence that increased torsional stress might lead to accelerated IDD.

## Figures and Tables

**Figure 1 ijms-22-13641-f001:**
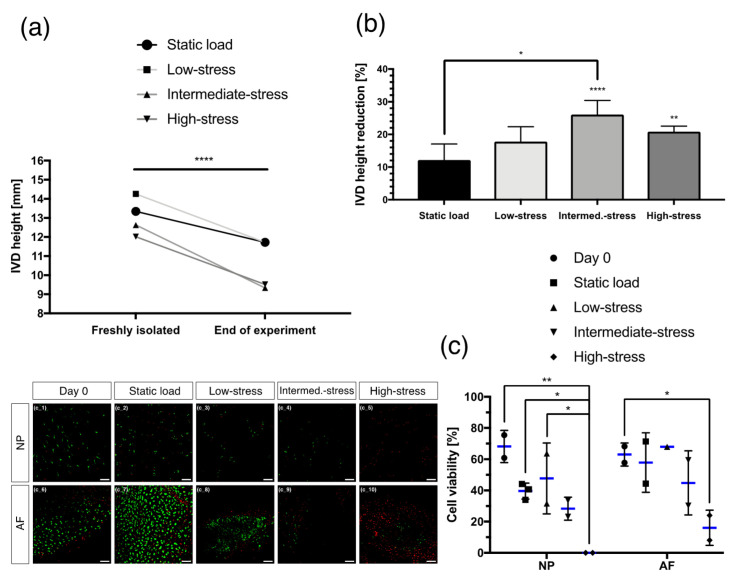
(**a**) Changes in the absolute IVD height under the influence of different loading regimes for seven days. Mean, N = 6–7. (**b**) Changes in the relative IVD height after seven days of loading with different loading regimes and compared to the initial IVD height after its dissection. Mean ± SD, N = 6–7. (**c**) Cell viability in the nucleus pulposus (NP) and in the annulus fibrosus (AF) at the end of each loading regime. Living cells were stained with calcein-AM (green) and dead cells with ethidium homodimer (red). Scale bar = 100 µm. Mean ± SD, n = 1–3. *p*-value: * < 0.05, ** < 0.01, **** < 0.0001.

**Figure 2 ijms-22-13641-f002:**
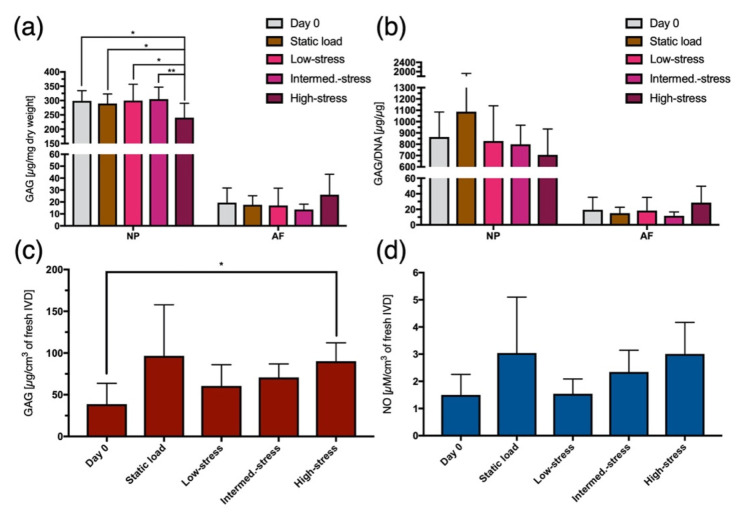
(**a**) GAG content in the nucleus pulposus (NP) and in the annulus fibrosus (AF) relative to their tissue dry weight. (**b**) GAG content in the NP and AF tissue relative to their DNA content. (**c**) Amount of GAG released from the IVDs into the culture medium in relation to the IVDs’ volume after their isolation from a bovine tail. (**d**) Amount of NO released from the IVDs into the culture medium in relation to the IVDs’ volume after their isolation from a bovine tail. Mean ± SD, N = 6–7, *p*-value: * <0.05, ** <0.01.

**Figure 3 ijms-22-13641-f003:**
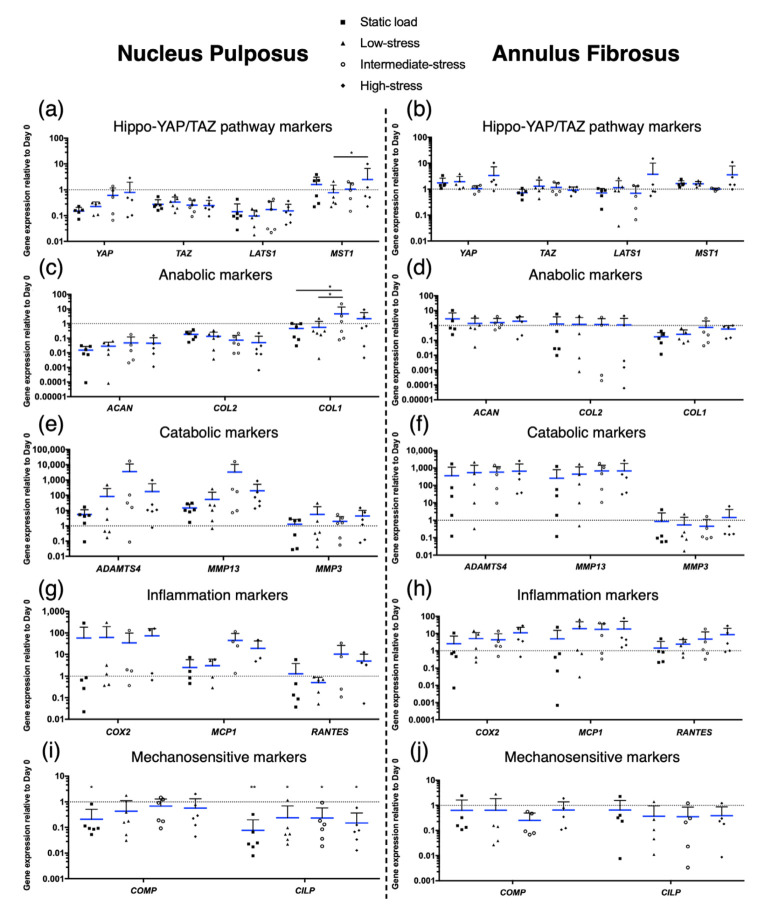
Relative gene expression of (**a**,**b**) key genes of the Hippo-YAP/TAZ pathway, including *YAP*, *TAZ*, *LATS1,* and *MST1*; (**c**,**d**) anabolic IVD-related extracellular matrix (ECM) genes, including *ACAN*, *COL2,* and *COL1*; (**e**,**f**) catabolic genes, including *ADAMTS4*, *MMP13,* and *MMP3*; (**g**,**h**) inflammatory genes, including *COX2*, *MCP1,* and *RANTES*; and (**i**,**j**) genes sensitive to mechanic stimuli, including *COMP* and *CILP*, in the nucleus pulposus (NP) and in the annulus fibrosus (AF). Mean ± SD, N = 3–6, *p*-value: * <0.05, ** <0.01.

**Figure 4 ijms-22-13641-f004:**
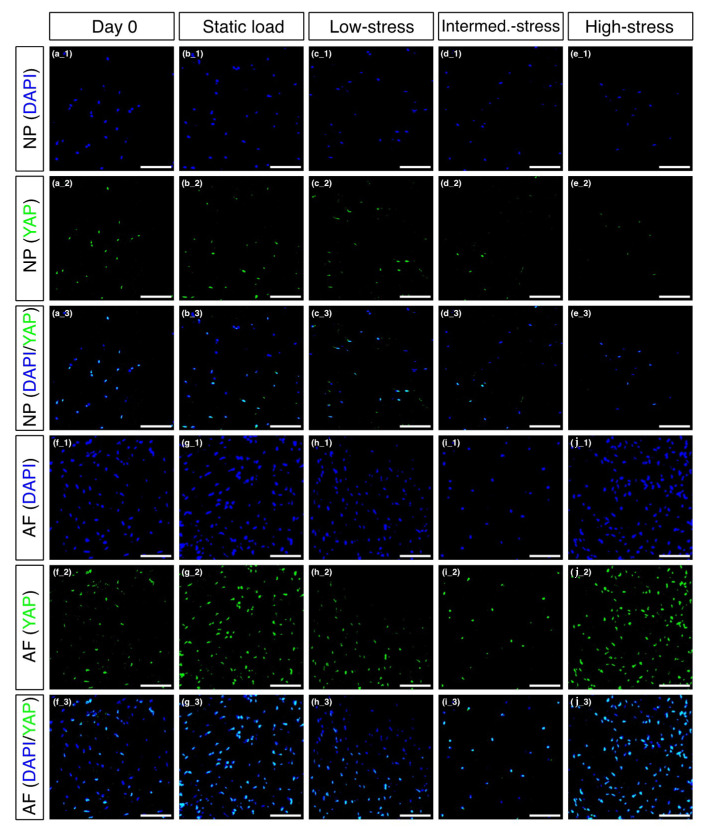
Immunofluorescent pictures of the nucleus pulposus (NP) and in the annulus fibrosus (AF). The sections were stained with DAPI (blue) and anti-YAP (green) and are shown individually as well as merged. (**a**) NP cells at day 0, (**b**) NP cells after static load, (**c**) NP cells after low-stress load, (**d**) NP cells after intermediate-stress load, (**e**) NP cells after high-stress load, (**f**) AF cells at day 0, (**g**) AF cells after static load, (**h**) AF cells after low-stress load, (**i**) AF cells after intermediate-stress load, (**j**) AF cells after high-stress load. Scale bar = 100 µm.

**Figure 5 ijms-22-13641-f005:**
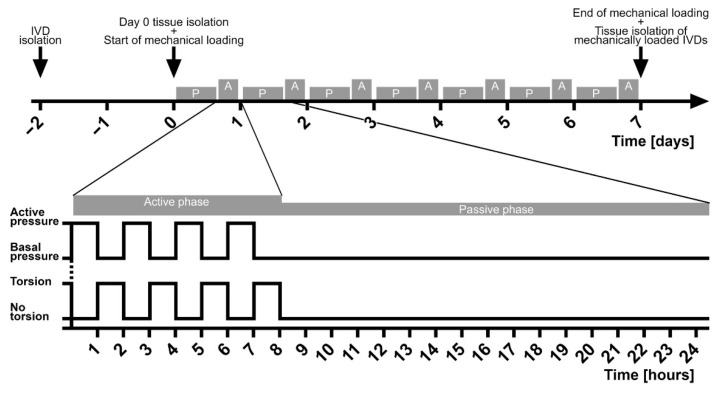
Experimental setup of the complex dynamic loading regimes.

**Table 1 ijms-22-13641-t001:** Mechanical loading regimes applied to the explanted IVDs. Day 0 was used as a reference for all other regimes; “static load” refers to the static mechanical loading regime, and “low-stress”, “intermediate-stress” as well as “high-stress” all refer to the complex dynamic mechanical loading regimes.

Regime	Description	
Day 0		No load. Prepared for further analysis after two days of free swelling.
Static load		Basal pressure of 0.1 MPa.
Low-stress	Passive phase	Basal pressure of 0.1 MPa.
	Active phase	Active pressure of 0.2 MPa alternates with a torsion of ±2° at a frequency of 0.05 Hz and a basal pressure of 0.1 MPa.
Intermediate-stress	Passive phase	Basal pressure of 0.1 MPa.
	Active phase	Active pressure of 0.4 MPa alternates with a torsion of ±8° at a frequency of 0.05 Hz and a basal pressure of 0.1 MPa.
High-stress	Passive phase	Basal pressure of 0.1 MPa.
	Active phase	Active pressure of 0.6 MPa alternates with a torsion of ±15° at a frequency of 0.05 Hz and a basal pressure of 0.1 MPa.

**Table 2 ijms-22-13641-t002:** Overview of all genes investigated and the primers used with qPCR in this study.

Gene Type	Full Name	Symbol	NCBI Gene ID	Forward and Reverse Primer Sequences
Reference gene	18S ribosomal RNA	*18S*	493779	f—ACG GAC AGG ATT GAC AGA TTG r—CCA GAG TCT CGT TCG TTA TCG
Anabolic markers	Aggrecan	*ACAN*	280985	f—GGC ATC GTG TTC CAT TAC AG r—ACT CGT CCT TGT CTC CAT AG
	Collagen Type 2, Alpha 1 Chain	*COL2*	407142	f—CGG GTG AAC GTG GAG AGA CA r—GTC CAG GGT TGC CAT TGG AG
	Collagen Type 1, Alpha 2 Chain	*COL1*	282188	f—GCC TCG CTC ACC AAC TTC r–AGT AAC CAC TGC TCC ATT CTG
Catabolic markers	ADAM Metallopeptidase with Thrombospondin Type 1 Motif 4	*ADAMTS4*	286806	f–AGA TTT GTG GAG ACT CTG r–ATA ACT GTC AGC AGG TAG
	Matrix Metallopeptidase 13	*MMP13*	281914	f–TCC TGG CTG GCT TCC TCT TC r–CCT CGG ACA AGT CTT CAG AAT CTC
	Matrix Metallopeptidase 3	*MMP3*	281309	f—CTT CCG ATT CTG CTG TTG CTA TG r—ATG GTG TCT TCC TTG TCC CTT G
Mechanosensitive markers	Cartilage Oligomeric Matrix Protein	*COMP*	281088	f—TGC GAC GAC GAC ATA CAC r—ATC TCC TAC ACC ATC ACC ATC
	Cartilage Intermediate Layer Protein	*CILP*	100336614	f—AGG ACT TCG TGC TGT ATG r—CTT GCT CAG GAG GTA GAC
Inflammatory markers	Cyclooxygenase 2	*COX2*	3283880	f—GGT AAT CCT ATA TGC TCT C r—GTA TCT TGA ACA CTG AAT G
	Monocyte Chemoattractant Protein 1	*MCP1*	281043	f—TCG CCT GCT GCT ATA CAT T r—TTG CTG CTG GTG ACT CTT
	Regulated Upon Activation, Normally T-Expressed, And Presumably Secreted	*RANTES*	327712	f—GTG CGA GAG TAC ATC AAC r—TTA GGA CAA GAG CGA GAA
Hippo-YAP/TAZ pathway markers	Yes1 Associated Transcriptional Regulator	*YAP*	100336629	f—CAG CAC AGC CAA TTC TCC AA r—TCC TGC TCC AGT GTT GGT AA
	Transcriptional Co-Activator With PDZ-Binding Motif	*TAZ*	614786	f—AGA TGG ACA CGG GAG AAA AC r—AGA AAA TCA GGG AAA CGG GT
	Large Tumor Suppressor Kinase 1	*LATS1*	535935	f—CAG CAG CTG CCA GAC CTA TTA r—TCC AGC TCT GTT TGC GGT TA
	Mammalian STE20-like Protein Kinase 1	*MST1*	514886	f—ATC ATG CAG CAA TGT GAC AG r—ATC AGA TAC AGA ACC AGC CC

## Data Availability

Data can be requested from the corresponding authors upon request.

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
