# Peer review of "In Situ Cell Signalling of the Hippo-YAP/TAZ Pathway in Reaction to Complex Dynamic Loading in an Intervertebral Disc Organ Culture"

_ijms, 2021, doi:10.3390/ijms222413641_

Round 1

Reviewer 1 Report

I reviewed the paper “In Situ Cell Signalling of the Hippo-YAP/TAZ Pathway in Reaction to Complex Dynamic Loading in an Intervertebral Disc Organ Culture” authored by A.S. Croft et al. I found the paper interesting that addressed an important issue the impact of Hippo-YAP/TAZ pathway in back pain. The followings are a few minor concerns that I would like the author to address:

  • The sample size was not mentioned in the methods. While statistical analysis was performed to identify the significant differences, the impact of the sample size to explore the true differences (Error Type II) was ignored.
  • How did the author measure the IVD height
  • How did the author evaluate the degeneration degree of the bovine tail IVD at the beginning o the experiment? Was it assumed that IVDs were healthy?
  • The limitation of the study, if any, is good to be addressed in the discussion.
  • Can 0.6 Mpa stress be considered as high stress for IVD?
  • Is it possible to see a correlation between the degree of IDD and subsequent inactivation of the Hippo-YAP/TAZ pathway if the sample size is increased or IVD from another species is used? Ovine IVD is a better animal model for LBP studies.

Author Response

Reviewer 1

Overall Consideration:

I reviewed the paper “In Situ Cell Signalling of the Hippo-YAP/TAZ Pathway in Reaction to Complex Dynamic Loading in an Intervertebral Disc Organ Culture” authored by A.S. Croft et al. I found the paper interesting that addressed an important issue the impact of Hippo-YAP/TAZ pathway in back pain. The followings are a few minor concerns that I would like the author to address:

R: We thank the reviewer for their time, valuable comments, and suggestions to improve our manuscript.

Q1: The sample size was not mentioned in the methods. While statistical analysis was performed to identify the significant differences, the impact of the sample size to explore the true differences (Error Type II) was ignored.

R: We apologize for any imprecise or missing description. For this study, we used up to seven biological replicates. However, the sample size varied slightly between the experiments. For this reason, we decided to always add the sample size at the end of each figure legend, indicated as N = “number of replicates”. Nevertheless, we agree with the reviewer that this should have been emphasized better in the Materials and Methods. Therefore, we amended the manuscript accordingly (page 14, line 493-495):

“Up to seven replicates were used for each experiment, however, the exact number of biological replicates = N and/or technical replicates = n are indicated in each figure legend.”

Furthermore, we revisited the power analysis of our particular statistical testing using ANOVA and an effective size difference of 0.8 (thus, looking for moderate to large differences) and a within-group sample size of N = 5, i.e., total sample size of 25. This resulted in a power of ~0.82 for this test, which is common sense for a type II error of ANOVA.

Q2: How did the author measure the IVD height?

R: Thank you for pointing this out. To measure the IVD height, we used a calliper and measured from one endplate to the other, making sure that the IVD’s endplates were aligned horizontally between the jaws of the calliper. We added to following section to the manuscript (page 10, line 349-352):

“As soon as five IVDs per bovine tail were isolated, the discs’ diameter was measured twice with a calliper at positions offset by 90°. Then the IVDs’ height was measured from one CEP to the other CEP, and it was ensured that the CEPs were aligned horizontally between the jaws of the calliper as previously described [56].”

Q3: How did the author evaluate the degeneration degree of the bovine tail IVD at the beginning o the experiment? Was it assumed that IVDs were healthy?

R: This is a very valid point. However, we are convinced that our day 0 samples were not degenerated for the following reasons: Firstly, the bovine tails were derived from relatively young cows, approximately one year old. Therefore, the IVDs should not show any significant evidence of disc degeneration, as disc degeneration is something more commonly seen in elderly subjects. Secondly, the cows were slaughtered the same morning the IVDs were isolated. We therefore know that the material we processed was very fresh and consequently hardly had any time to degenerate. Thirdly, great care was taken during the isolation process to avoid any kind of IVD trauma. IVDs that were damaged during the isolation process were immediately excluded from the study. Finally, all genes were expressed relative to day 0. That is, if day 0 had already degenerated, then we would not have seen any upregulation of the inflammatory and catabolic markers, at least not to this extent.

Q4: The limitation of the study, if any, is good to be addressed in the discussion.

R: We agree with the reviewer that the limitations are an important element of any study, and that they should be mentioned in the discussion. We would therefore kindly like to point out that in the initial submission we already devoted a paragraph to the limitations in the discussion (page 10, line 330-338).

The paragraph reads as follows:

“Finally, there are limitations to this study that need to be addressed. Firstly, as previously discussed, there was no loading regime that used a sine wave function to control the compression. Therefore, we cannot state how the IVDs and more specifically the Hippo-YAP/TAZ pathway would have reacted if the samples had been exposed to a constantly varying pressure. Secondly, only activated / non-phosphorylated YAP was evaluated for the immunohistochemistry assay. The addition of an antibody specifically for phosphorylated YAP and its subsequent visualization would have made it possible to determine the ratio between activated and deactivated YAP per cell and consequently what percentage of the total YAP translocated into the cell’s nucleus.”

Q5: Can 0.6 Mpa stress be considered as high stress for IVD?

R: We thank the reviewer for pointing this out. According to a study by Wilke et al. an intradiscal pressure of approximately 0.6 MPa is reached by an adult in a relaxed standing position (Wilke et al. Spine 1999). However, here we were restricted by the mechanical capabilities of our custom-built bioreactor, which only allows forces up to 150 N, which corresponds to little more than 0.6 MPa depending on the IVD’s size. Nevertheless, we would like to emphasize that the high-stress regime not only consisted of 0.6 MPa pressure, but also included a torsion of 15°. Torsion angles greater than 5° are already in the supraphysiological range and consequently harm the IVD (Chan et al. PLoS One 2013), and angles greater than 15° have even been reported to provoke complete failure of the IVD (Farfan et al. J Bone Jount Surg Am 1970). Therefore, we do think that the high-stress regime is justified.

Q6: Is it possible to see a correlation between the degree of IDD and subsequent inactivation of the Hippo-YAP/TAZ pathway if the sample size is increased or IVD from another species is used? Ovine IVD is a better animal model for LBP studies.

R: Based on our data, we could not see a clear correlation between the degree of IDD and subsequent inactivation of the Hippo-YAP/TAZ pathway using a sample size of 5-6 biological replicates. For the NP, we do not expect to see a significant correlation between the degree of IDD and subsequent inactivation of the Hippo-YAP-TAZ pathway with increased samples size, because the relative gene expression of YAP and TAZ between day 0 and the high-stress loading regime revealed p-values of 0.997 and 0.687, respectively. For the AF, we got a p-value of > 0.999 for TAZ and p = 0.332 for YAP, when day 0 and the high-stress loading regime were compared. Hence, it is possible that the relative expression of YAPcould become significant with a notably increased sample size in the AF. However, this seems very speculative, as the p-value could just as well increase with a bigger sample size.

Regarding the reviewer’s comment on whether ovine IVDs are a better model to study LBP, we do think that they are both legitimate models for LBP and IDD studies. Both IVD types have a similar cell density and a comparable composition of their extracellular matrix, and in both species the NP notochordal cells are not retained throughout their life, which is also the case in humans (Daly et al. Biomed Res Int. 2016). Furthermore, ovine and bovine IVDs are both well established and frequently used models for studying IDD (Ovine: Illien-Jünger et al. Spine 2010 and Veres et al. Eur Spine J 2010; Bovine: Chan et al. PLoS One 2013 and Zhou et al. J Orthop Translat 2020).

Reviewer 2 Report

Overall Consideration:

In the manuscript, “In Situ Cell Signaling of the Hippo-YAP/TAZ Pathway in Reaction to Complex Dynamic Loading in Intervertebral Disc Organ Culture”, the authors utilized different loading regimes to test their effects on intervertebral disc height, cell viability within the disc, matrix content, gene expression, and phosphorylation of the mechanosensitive protein YAP.  Although the study is fairly descriptive, it does represent a valuable contribution to the field.  The manuscript it well written and easy to follow.  The data are presented well, with biological replicates shown.  A few minor suggestions are listed below:

  • Quantification of the percent of cells with nuclear staining would enhance the manuscript
  • The authors should spell out the abbreviation for YAP/TAZ in the first instance of use within the introduction
  • There is a typo on line 161: upregulated is missing the “d”

Author Response

Reviewer 2

Overall Consideration:

In the manuscript, “In Situ Cell Signalling of the Hippo-YAP/TAZ Pathway in Reaction to Complex Dynamic Loading in Intervertebral Disc Organ Culture”, the authors utilized different loading regimes to test their effects on intervertebral disc height, cell viability within the disc, matrix content, gene expression, and phosphorylation of the mechanosensitive protein YAP.  Although the study is fairly descriptive, it does represent a valuable contribution to the field.  The manuscript it well written and easy to follow.  The data are presented well, with biological replicates shown.  A few minor suggestions are listed below:

R: We thank you very much for the positive review and your suggestions to further improve our manuscript.

Q1: Quantification of the percent of cells with nuclear staining would enhance the manuscript

R: We agree with the reviewer that the quantification of YAP-positive nuclei would enhance the quality of the manuscript. Therefore, we counted the YAP-positive nuclei and adapted the manuscript as follows:

Revised section in the Results (page 5, line 192-199):

“Finally, the amount YAP-positive nuclei relative to the number of nuclei per regime was very similar throughout the NP tissue (day 0: 61.3 %, static load: 60.5 %, low-stress: 55.6 %, intermediate-stress: 56 % and high-stress: 57.1 % YAP-positive, respectively; Figure 4a_2 until Figure 4e_2) but underrepresented in the AF’s day 0 (50 % YAP-positive; Figure 4f_3) and low-stress regime (47.9 % YAP-positive; Figure 4h_3) compared to the other regimes in the AF (static load: 78.9 %, intermediate-stress: 67.9 % and high-stress: 77 % YAP-positive, respectively)”

Revised section in the Materials and Methods (page 14, line 468-470):

“Finally, YAP-positive nuclei were counted manually using the ImageJ cell counter plugin (version 1.52q for Mac OS X).”

Q2: The authors should spell out the abbreviation for YAP/TAZ in the first instance of use within the introduction

R:  Thank you for pointing this out. In the revised version, the abbreviations for YAP/TAZ are now spelled out in the first instance of use within the introduction (page 2, line 55-58).

The revised section now reads as follows:

“The Hippo-YAP/TAZ pathway (yes-associated protein 1 and transcriptional coactivator with PDZ-binding motif, respectively) is an evolutionarily conserved signalling cascade that was first discovered in Drosophila melanogaster and later confirmed in mammals [17,18].”

Q3: There is a typo on line 161: upregulated is missing the “d”

R: Thank you for this notification. We have corrected the respective typo (page 5, line 168).
